# The Novel Antipsychotic Lumateperone (Iti-007) in the Treatment of Schizophrenia: A Systematic Review

**DOI:** 10.3390/brainsci13121641

**Published:** 2023-11-26

**Authors:** Giulio Longo, Angelica Cicolini, Laura Orsolini, Umberto Volpe

**Affiliations:** Unit of Clinical Psychiatry, Department of Neurosciences/DIMSC, School of Medicine, Polytechnic University of Marche, Via Conca 71, 60126 Ancona, Italy; giulio.longo1996@gmail.com (G.L.); angelica-cicolini@outlook.it (A.C.); u.volpe@staff.univpm.it (U.V.)

**Keywords:** ITI-007, ITI-722, lumateperone, novel antipsychotics, schizophrenia, antipsychotics

## Abstract

Lumateperone (also known as ITI-007 or ITI-722) represents a novel second-generation medication characterized by a favorable safety and tolerability profile. This is attributed to its notable selectivity for D2 receptors within specific regions of the brain. The U.S. Food and Drug Administration (FDA) granted approval for the treatment of schizophrenia in adults in December 2019. Additionally, it gained approval for addressing depressive episodes associated with bipolar I and II disorders in adults, either as a standalone therapy or in conjunction with lithium or valproate, in December 2021. The objective of this investigation is to systematically review the existing literature to assess the safety, tolerability, and efficacy of lumateperone in the treatment of schizophrenia. Lumateperone has demonstrated effectiveness in addressing positive, negative, and cognitive symptoms associated with schizophrenia. The evaluation of safety indicators in the reviewed studies indicates that lumateperone is deemed to be a well-tolerated and safe antipsychotic. Additional research is warranted to explore lumateperone’s efficacy in managing major depressive disorders, behavioral issues in Alzheimer’s disease and dementia, sleep maintenance insomnia, bipolar disorders, and personality disorders.

## 1. Introduction

Lumateperone (ITI-007 or ITI-722, 1-(4-fluorophenyl)-4-(4-methyl-1,4,12triazatetracyclo [7.6.1.05,16.010,15] hexadeca-5,7,9(16)-trien-12-yl) butan-1-one) is a novel second-generation antipsychotic that exerts its activity as a strong serotonin 5-HT2A receptor antagonist by inhibiting the serotonin transporter (SERT) and acting as a partial agonist-antagonist at D2 receptors [1,2,3]. Lumateperone was developed by Intra-Cellular Therapies, Inc. (New York, NY, USA) under a license from Bristol-Myers Squibb [4]. Lumateperone is safeguarded by a composition-of-matter patent set to expire in December 2028 [5]. The Food and Drug Administration (FDA) granted approval for lumateperone (marketed as Caplyta^®^) in December 2019 for the treatment of schizophrenia in adults [5]. Subsequently, in December 2021, lumateperone received approval for use in adults with bipolar depression, either as a standalone therapy or in conjunction with lithium or valproate [6]. Currently, lumateperone is undergoing phase II clinical development for addressing sleep maintenance insomnia [7] and is in phase III clinical development for the treatment of major depressive disorder [8]. Additionally, the exploration of lumateperone’s potential extends to the management of behavioral disorders in individuals with dementia, as well as those affected by Alzheimer’s disease [9,10].

### 1.1. Chemistry

Lumateperone (CAS number 313368-91-1), also known as ITI-007 or ITI-722, is marked as a tosylate salt (molecular formula: c24h28fn3o). Its molecular weight is 393.5 g/mol [1]. Figure 1 shows lumateperone’s chemical structure [4].

### 1.2. Pharmacokinetic Profile

Lumateperone exhibits a heightened oral bioavailability and favorable absorption characteristics [7]. Rapid absorption in the gastrointestinal tract results in a peak concentration (Cmax) within 1 to 2 h and a mean time of maximum concentration (Tmax) of 3 to 4 h following the once-daily oral administration of a 42 mg capsule [11]. The intake of a high-fat meal is associated with an estimated 33% reduction in Cmax, accompanied by a 9% increase in the area under the concentration–time curve (AUC) [5]. In the presence of food, the mean Tmax is delayed by 1 h compared to a fasted state [2]. Steady-state concentrations are achieved in approximately 5 days, and the drug’s half-life ranges from 13 to 21 h [7]. Intravenous administration yields a volume of distribution of about 4.1 L/kg, and the drug exhibits a human-plasma-protein-binding rate of 97.4% [5]. Despite being highly lipophilic at pH 7.4, this compound is characterized by high permeability and bidirectional permeability via Multidrug Resistance Protein 1 (MDR1). These properties enable the drug to traverse both the small intestine and the blood–brain barrier [12].

Lumateperone is significantly metabolized by numerous enzymatic pathways, primarily by uridine 5′-diphospho-glucuronosyltransferases (UDP-glucuronosyltransferase, UGT), aldoketoreductase, and cytochrome P450 [1]. The primary metabolite of lumateperone, ICI200131, results from the reduction of the carbonyl side-chain via ketone reductase, forming an alcohol compound [12]. In the liver, CYP3A4 determines the formation of the last two active metabolites, IC200161 and IC200565, through dealkylation [1]. These metabolites share similar mechanisms with lumateperone, leading to an extension of the compound’s action [1]. Because lumateperone is a CYP3A4 substrate, drug–drug interactions are possible. For patients who also take a CYP3A4 inducer (e.g., carbamazepine), lumateperone is suggested to be avoided, because the associated administration will reduce the exposure of lumateperone itself [1]. In addition, the coadministration of lumateperone with moderate or strong CY3A4 inhibitors (e.g., fluvoxamine) should be avoided, as increased exposure to lumateperone may augment the probability of toxicity [11]. Moreover, the administration of lumateperone with UDP-glucuronosyltransferase (UGT) inhibitors (e.g., valproic acid) results in the same effect as coadministration with CY3A4 inhibitors [2]. When taken orally, approximately 58% of radiolabeled lumateperone is excreted in the urine, with 29% being excreted in the feces [2]. Less than 1% of the dose is excreted unchanged in the urine [2]. Lumateperone metabolites are water soluble and are totally excreted in the urine [13].

### 1.3. Pharmacodynamic Profile

Lumateperone, classified as a second-generation antipsychotic (SGA) drug, functions as a selective and simultaneous modulator of serotonin, dopamine, and glutamate, all of which are associated with severe mental illness (SMI) [14]. Similar to other SGAs, lumateperone acts as an antagonist at 5-HT2A and D2 receptors. It exhibits a high affinity for 5-HT2A, a moderate affinity for D2, and a low affinity for α-1 and histamine-1 receptors (H1) [13]. Its reduced binding affinity for α-1 and H1 receptors is linked to a lower incidence of adverse events compared to other SGAs such as aripiprazole, quetiapine, and olanzapine [5]. Lumateperone functions as a dopamine receptor phosphoprotein modulator (DPPM) [15], serving as both a presynaptic partial agonist and a postsynaptic antagonist at D2 receptors [16]. This distinctive mechanism leads to a reduction in presynaptic dopamine release and results in the blockade of postsynaptic dopaminergic activity, leading to a significant decrease in dopaminergic signaling [16,17]. Aripiprazole and its derivatives are the only other second-generation antipsychotic (SGA) drugs that employ this “dual-method” approach of D2 presynaptic agonism and postsynaptic antagonism [18]. Lumateperone has a high affinity for D2 receptors only in mesolimbic and mesocortical regions (Ki = 32 nM) and has a lower affinity for the D2 receptors within the nigrostriatal pathways [15,19]. This activity is responsible for its antipsychotic efficacy without the risk of onset of extrapyramidal adverse events or hyperprolactinemia [3]. Furthermore, there is a dose-dependent relationship in the level of D2 receptors occupancy by lumateperone and, hence, in its clinical activity [20]. A 10 mg dose of lumateperone results in minimal occupancy (around 12%) of striatal dopamine D2 receptors, whereas a 40 mg dose of lumateperone can lead to occupancy as high as 39% of striatal dopamine D2 receptors [20].

ITI-007 triggers the phosphorylation of glycogen synthase kinase 3β (GSK-3β) in the mesolimbic/mesocortical dopamine systems [13,16], leading to the inhibition of kinase activity [21]. An anomalous activation of GSK3β signaling pathways is associated with the etiology of psychiatric diseases, while kinase inhibition has been suggested to have therapeutic benefits in schizophrenia and bipolar disease [22,23]. In addition, lumateperone’s partial agonism on D1 receptor induces an augmented phosphorylation of GluN2B subunit of glutamatergic N-methyl-d-aspartate (NMDA) receptors in mesolimbic brain region [16]. D1 receptors’ partial agonism also facilitates the activation of NMDA and AMPA receptors in the prefrontal cortex [24]. Therefore, increased glutamatergic neurotransmission in NMDA channels may contribute to the lumateperone activity on the cognitive impairments and negative symptoms, as NMDA receptors’ activity is deficient in schizophrenic patients [25]. Moreover, lumateperone acts as a strong serotonin 5-HT2A receptor antagonist (Ki = 0.54 nM) and has a 60-fold higher affinity for 5-HT2A receptors than for D2 receptors, which may be associated with the relatively low risk of extrapyramidal symptoms (EPS) reported in clinical trials [11,17,26]. Finally, lumateperone acts as a serotonin reuptake transporter (SERT) inhibitor (Ki = 62 nM), determining antidepressant effects and potentially attenuating the negative symptoms of schizophrenia [13].

### 1.4. Preclinical Studies

Preclinical studies examined the neurobiology of lumateperone, primarily related to its neuropharmacology and toxicological profile [16,27].

The in vivo and in vitro studies have demonstrated that lumateperone is a strong serotonin 2A receptor antagonist, a presynaptic partial agonist and postsynaptic antagonist of dopamine receptor 2 (D2), a modulation of glutamatergic transmission mediated by the dopaminergic D1 receptor, and a serotonin reuptake inhibitor [27]. Like other atypical antipsychotic medications, lumateperone exhibits high-affinity binding (nM concentrations) to human serotonin 5HT2A and dopamine D2 receptors [16]. While variations were observed in different dosing regimens, lumateperone exhibits higher affinity binding to the 5HT2A receptor (Ki = 0.52 to 10 nM) than to the dopamine D2 receptor (Ki = 19.2 to 32 nM). Additionally, it demonstrates moderate to high affinity for dopamine D1 receptors (20 to 78 nM), D4 receptors (39.7 to 104 nM), and α 1b adrenergic receptors (36.9 nM), all of which are implicated in the effectiveness of certain antipsychotic drugs [16]. Furthermore, lumateperone binds with high affinity to the serotonin transporter (SERT; Ki = 16 to 33 nM) [16]. The higher binding affinity for 5HT2A receptors than for D1 and D2 receptors allows for the complete saturation of cortical 5HT2A receptors and antipsychotic efficacy at doses that do not induce motor side effects, stemming from the complete occupancy of striatal D2 receptors [27]. Lumateperone undergoes extensive metabolism, resulting in multiple pharmacologically active metabolites in both humans and non-human species [16]. These metabolites can be detected in the systemic circulation at levels similar to or higher than lumateperone, potentially contributing to its overall pharmacological activity. Notably, the desmethyl-metabolite IC20161 and the reduced carbonyl-metabolite IC200131 exhibit binding profiles comparable to those of lumateperone [27]. Lumateperone and its metabolites do not display relevant binding affinity to D3, 5-HT1A, and 5-HT7 receptors, which are targets contributing to the efficacy of other atypical antipsychotic agents [27]. To evaluate the toxicological profile of lumateperone, studies were conducted in mice, rats, and dogs, involving oral administration for 3, 6, and 9 months. Additionally, investigations into its carcinogenic potential were carried out via oral administration for up to 21 months in rats and mice. Oral administration of lumateperone resulted in systemic intracytoplasmic aggregation of pigmented material in dogs, rats, and mice. The distribution and quantity of this pigment accumulation appeared to increase with higher doses and longer treatment durations [27]. Of particular interest is the accumulation of pigmented material in tissues with limited regenerative capacity. Pigment accumulation was observed in the brain and spinal cord in all three species, in cardiomyocytes, and in the retina of rats. However, recovery was assessed in a reasonably short period (1 to 2 months). In rats, the aggregation of pigmented material was evident on coarse necropsy and frequently localized to parenchymal cells, macrophages, and other mononuclear inflammatory cells in several organs, eventually accumulating in extracellular spaces (e.g., fibrous connective tissue displacing myocardiocytes in the heart and accumulating in the pulmonary interstitium). The aggregation of pigmented material was associated with adverse effects in the central nervous system (e.g., histiocytic inflammation in the brain and axonal degeneration in the spinal cord), peripheral nervous system (e.g., axonal degeneration in peripheral nerves), eye (retinal degeneration), and heart (cardiomyopathy) [16,28]. The nature of the constituents representing the intracellular pigmented material identified in toxicology studies has not been fully characterized. Nevertheless, it does not appear to be a common endogenous intracytoplasmic pigment such as lipofuscin or hemosiderin [28]. Based on in vitro assays, the intracellular pigmented material is likely formed by polymers and/or protein agglomerates made up of aniline metabolites of lumateperone, IC201337, and IC201338 [27]. The physiological and clinical relevance of the storage of pigmented materials are not entirely clear, but if the pigmented materials originated from the aniline metabolites, they probably cause lysosomal dysfunction [29,30]. In fact, the aniline metabolites are cationic amphiphilic amines (CADs) [31]. At physiological pH, CADs are not ionized and easily diffuse across lipid bilayers. When CADs enter a cellular organelle with an acidic pH, such as lysosomes, they are ionized, resulting in intralysosomal accumulation and subsequent cellular dysfunction [27,31]. It is important to underline that the aniline metabolites of lumateperone, IC201337 and IC201338, which seem to be responsible for the aggregation of pigmented material, were not detected in humans at measurable levels. However, the possibility for lumateperone or other metabolites to accumulate in lysosomes and cause the observed in vivo and in vitro toxicities cannot be excluded [27]. The toxicological potential of such accumulations is similar to that of other antipsychotic agents known to accumulate in lysosomes (e.g., cariprazine and aripiprazole) [32,33]. The NOAEL (No Observed Adverse Effect Level) for general toxicity in rats was 2.4 times the maximum recommended human daily dose (MRHD) of 42 mg on a mg/m^2^ basis; in dogs, it was 2 times the MRHD of 42 mg on a mg/m^2^ basis< and in mice, it was approximately 2.4 times the MRHD of 42 mg lumateperone on a mg/m^2^ basis. The LD50 was not determined [27].

### 1.5. Posology and Pharmacological Formulation

Lumateperone is administered orally in a capsule that contains the drug formulated as a crystalline, tosylate salt. Visibly, the pill is half white and half blue. On the white part there is printed "ITI-007 42 mg” printed in white [6]. The approved dosage according to the FDA is 42 mg, to be taken once daily, preferably at bedtime [5]. Dosages of 120 mg/day have been shown not to result in statistically significant improvements in efficacy [26]. Lumateperone 20 mg is equivalent to 14 mg active moiety, lumateperone 40 mg tosylate is equivalent to 28 mg active moiety, lumateperone 60 mg tosylate is equivalent to 42 mg active moiety, and lumateperone 120 mg tosylate is equivalent to 84 mg active moiety [14].

### 1.6. Pregnancy and Breastfeeding

There is no available information about the clinical use of lumateperone during pregnancy and breastfeeding. Lumateperone is 97.5% bound to plasma proteins; thus, its presence in milk is probably low. However, the manufacturer recommends discontinuing lumateperone during breastfeeding. Until more safety data become available, an alternate drug may be preferred [34].

### 1.7. Aims of the Study

The aim of the study is to conduct a systematic literature review of the existing evidence regarding the safety, tolerability, and effectiveness of lumateperone in the management of schizophrenia, incorporating the most recent published articles to ensure the information is as up to date as possible.

## 2. Materials and Methods

A systematic review protocol was developed and registered online with PROSPERO (CRD42023479279). According to the PRISMA guidelines [35], a systematic English literature review was performed by consulting the PubMed/MEDLINE database, from inception until 05 January 2023. The adopted PubMed strings were: “(*Lumateperone* [Title/Abstract]) AND *Schizophrenia* [Title/Abstract]”; “(*ITI-007* [Title/Abstract]) AND *Schizophrenia* [Title/Abstract]”; “(*ITI-722* [Title/Abstract]) AND *Schizophrenia* [Title/Abstract]”; “(*Lumateperone* [Title/Abstract]) AND *Psychiatric Disorders* [Title/Abstract]”; “(*Lumateperone* [Title/Abstract]) AND *Psychosis* [Title/Abstract]”. In addition, the websites https://www.clinicaltrials.gov (last access on 5 January 2023), https//intracellulartherapies.com (last access on 5 January 2023), and https://www.accessdata.fda.gov (last access on 5 January 2023) were searched to include studies not yet published. Identified studies were independently reviewed for eligibility by two authors (G.L. and A.C.) in a two-step-based process; a first screening was performed based on title and abstract, while the full texts were retrieved for the second screening. At both stages, disagreements put forward by the reviewers were resolved by consensus between the screening authors (G.L. and A.C.). Then, the data were extracted by two authors (G.L. and A.C.) and disagreements were resolved by a third author (L.O.) using an ad hoc developed data extraction spreadsheet. With the initial set of keywords, by integrating all databases, 62 studies were identified. After removing duplicates, only 48 articles were selected. Of these, 11 relevant studies were finally included.

## 3. Results

Figure 2 presents the PRISMA flow diagram, illustrating the process of reviewing the articles identified through the source search. Table 1 provides a summary of all concluded clinical trials that have assessed the safety, tolerability, efficacy, and effectiveness of lumateperone in the treatment of individuals diagnosed with schizophrenia.

### 3.1. Phase I Clinical Trials

An open-label, non-randomized phase I clinical trial (ITI-007-025; NCT04709224), was carried out to assess the tolerability, safety and pharmacokinetics of progressively increasing the single dose of a subcutaneous long-acting injectable (LAI) formulation of lumateperone. The study was conducted on 24 adult patients (aged 18–50 years) diagnosed with schizophrenia, who had been clinically stable and free from acute exacerbations of psychosis for at least three months prior to the screening, as determined by the investigators [36]. These patients had been on a consistent dose of antipsychotic medication, including lumateperone, for a minimum of three months before the screening visit. All patients received oral lumateperone for 5 days, followed by a 5-day washout of lumateperone and then by a single dose of lumateperone LAI. In this trial, there were four study arms: cohort 1 received a subcutaneous injection of 50 mg LAI lumateperone on the abdomen, cohort 2 received 100 mg on the abdomen, cohort 3 received 200 mg on the abdomen, and cohort 4 received 100 or 200 mg on the outer area of the upper arm. The primary endpoints were pharmacokinetics measures: Cmax, Tmax, AUC, and terminal elimination half-life [36]. Until now, no data are available regarding the outcomes of this clinical trial.

Another phase I 5-day open-label clinical trial (ITI-007-020; NCT04779177) was carried out to evaluate the tolerability, safety and pharmacokinetics of a single oral dose of lumateperone on patients around 13 and 17 years old affected by schizophrenia or schizoaffective disorder clinically stable and free from acute exacerbation of psychosis for at least 3 months before the screening. In this trial there, were two study arms: one group received a daily dose of lumateperone 42 mg for a duration of 5 days, and the other group received a daily dose of lumateperone 28 mg for a duration of 5 days. The primary endpoints were pharmacokinetics measures: Cmax, Tmax, apparent oral clearance (CL/F), terminal elimination half-life (T½) and AUC [37]. Until now, no data are available regarding the outcomes of this clinical trial.

### 3.2. Phase II Clinical Trials

A safety/efficacy phase II double-blind, placebo-controlled, multi-center study (ITI-007-005; NCT01499563) randomized 335 adult patients (aged 18–55 years) with acute reacutization of schizophrenia to lumateperone 60 mg once daily (42 mg of active moiety; *n* = 76), lumateperone 120 mg once daily (84 mg of active moiety; *n* = 80), risperidone 4 mg daily (active control; *n* = 75) or placebo (*n* = 80) for 4 weeks [26]. The primary outcome was a modification in the Positive and Negative Syndrome Scale (PANSS) total score from the initial assessment to day 28. The secondary endpoint included variations in PANSS subscales during the same period. Notably, a significant distinction emerged in the PANSS total score change between the 60 mg lumateperone group and the placebo group (Least-Squares Mean Change (LSMD) −13.2 vs. −7.4 for placebo; *p* = 0.01). A statistically significant difference in the PANSS total score was also observed between placebo and 4 mg risperidone (LSMD −13.4 vs. −7.4 for placebo; *p* = 0.013) [26]. There was no significant variation in the PANSS total score with 120 mg lumateperone; however, despite improvement in positive symptoms, the distinctions were not statistically significant when compared to the placebo [26]. The exact reason for the insufficient efficacy of the 120 mg dose remains uncertain. This outcome could be ascribed to variances in the participants, potential measurement inaccuracies, or the hypothesis that the combined effect of presynaptic dopamine D2 agonism and postsynaptic D2 antagonism could attenuate the effectiveness of the elevated dose [1,26].

In contrast to the placebo, both 60 mg of lumateperone and 4 mg of risperidone demonstrate significant improvement in PANSS positive symptoms and PANSS general psychopathology [26]. Negative symptoms improved only with 60 mg of lumateperone; nevertheless, this was not clinically relevant. Both risperidone and a 120 mg dose of lumateperone demonstrated no improvement in negative symptoms [26]. The persistence of the negative symptoms could be explained by the low baseline negative symptomatology [14]. For patients exhibiting significant negative symptoms at the initial assessment, the administration of lumateperone at a 60 mg dose resulted in a decrease in the intensity of negative symptoms, as assessed by the PANSS negative symptom subscale. Approximately one-third of the patients fulfilled the criteria for marked negative symptoms at baseline and were included in a predefined exploratory subgroup analysis. Within this specific subgroup, the use of 60 mg of lumateperone led to a reduction in symptom severity, as indicated by the PANSS negative symptom subscale (LSMD −3.0 points vs. −1.3 points (placebo), effect size = −0.34). In contrast, risperidone showed a minimal increase in negative symptoms compared to the placebo (LSMD + 1.2 points vs. −1.3 points (placebo), effect size = +0.02) [26].

In addition, the safety data for lumateperone in the ITI-007-005 study reported that patients in the 60 mg and 120 mg lumateperone groups did not experience any severe treatment-emergent adverse events (TEAEs) [2]. The relative risk of TEAEs for 60 mg lumateperone is 1.14, exhibiting no significant change compared to placebo (*p* = 0.346). In contrast, the relative risk for 120 mg lumateperone was 1.3 (*p* = 0.024). The predominant adverse event was somnolence and sedation, reported in 13% of the placebo group, 17% of the lumateperone 60 mg group, 21% of the risperidone 4 mg group, and 32.5% of the lumateperone 120 mg group [2,26]. The administration of lumateperone did not show any correlation with increased EPS, electrocardiogram (ECG) changes, or metabolic alterations. Both doses of lumateperone demonstrated lower weight gain compared to placebo and significantly reduced levels of prolactin, fasting glucose, total cholesterol, and triglycerides in comparison to risperidone [26].

Another phase II, 6-week open-label efficacy and safety study of lumateperone (ITI-007-303; NCT03817528) was carried out in adults with clinically stable schizophrenia [17,38]. In the initial phase of the investigation, 302 adult patients (ages 18–60), diagnosed with schizophrenia and medically stable, underwent a transition from their standard of care (SOC) antipsychotic therapy to a six-week regimen of 60 mg daily lumateperone. Subsequently, the patients were reverted to their original SOC or another approved antipsychotic for a two-week period [17,38]. The primary objective was to evaluate TEAEs, vital signs, laboratory tests, and the presence of EPS. The assessment of schizophrenia symptoms was conducted through the PANSS total score. TEAEs occurred in 45.5% of the patients, with the most common drug-related TEAEs being somnolence (6.6%), dry mouth (5.3%) and headache (5.3%). The majority of TEAEs were of mild or moderate severity, and adverse events related to EPS were infrequent (1.0%). Five patients had TEAEs of serious severity, including migraine, panic attack, hepatic enzyme augmentation, and non-cardiac chest pain. Patients experienced statistically significant improvements in metabolic parameters, endocrine parameters and weight; however, these gains were reversed when patients transitioned back to their previous maintenance antipsychotic [17,38]. In the course of this study, the shift from prior antipsychotics to a six-week lumateperone regimen resulted in patients maintaining stability or continued improvement in symptomatology, as evidenced by mean improvements in the PANSS total score (mean change from baseline at day 42–2.2 [95% CI = −3.2 to −1.2]) [17,38]. A second part open-label, long-term study was undertaken to assess the safety and effectiveness of lumateperone 60 mg in individuals with clinically stable schizophrenia. In this study, 603 patients transitioned from their SOC to a once-daily dose of 60 mg lumateperone for a duration of up to 1 year [39]. A preliminary analysis conducted at the one-year mark in the long-term study revealed that out of 603 patients in the safety population, 84 individuals (13.9%) discontinued the study medication due to one or more TEAEs. The adverse event that most usually caused the suspension of medication was the exacerbation of schizophrenia (32 patients; 5.3%). Other TEAEs which resulted in the exit from the clinical study for more than one patient were anxiety, suicidal ideation, headache, extrapyramidal disorder, dizziness, diarrhea, nausea, vomiting, somnolence, insomnia and fatigue. The majority of these adverse events exhibited mild or moderate intensity [39]. Various metabolic parameters, as well as mean prolactin levels, decreased from the SOC baseline, including changes in body weight and BMI. Treatment with lumateperone 60 mg was associated with significant reductions in PANSS total score from baseline, and this improvement persisted throughout the entire study duration. Among patients with moderate-to-severe depression symptoms at baseline (Calgary Depression Scale for Schizophrenia CDSS > 5), the average CDSS scores diminished from 7.4 (baseline) to 3.1 (day 300); 60% of patients met the CDSS response criteria (50% improvement from baseline) by day 75, and this response rate was maintained through day 300 [39].

An additional open-label phase II study (ITI-007-008; NCT02288845) was carried out to investigate the correlation between lumateperone dosage, plasma concentrations, and brain D2 receptor occupancy (D2RO) in individuals with clinically stable schizophrenia [17]. A cohort of 14 patients, aged 18 to 60, all diagnosed with stable schizophrenia, participated in this study. Out of these, 10 patients received a daily 60 mg dose of lumateperone for a duration of two weeks following a two-week washout period, and successfully completed the study. The study’s primary focus was the assessment of D2 receptor occupancy (D2RO) using positron emission tomography (PET). The study shows a mean peak D2RO in dorsal striatal of about 40% at 60 mg lumateperone. This occupancy level is lower than what is typically observed with many other antipsychotic medications administered at their effective doses. This is likely a contributing factor to the positive safety and tolerability characteristics associated with lumateperone. [13,15,16]. In this study, lumateperone demonstrated good tolerability and a favorable safety profile. There were no statistically significant alterations in vital signs, electrocardiograms (ECGs), or clinical chemistry laboratory values, and no elevation in prolactin levels was observed. The most frequently reported adverse events (occurring in more than two patients) were mild-to-moderate headaches (4/10, 40%) and mild sedation (4/10, 40%). No adverse events associated with akathisia or other extrapyramidal symptoms (EPS) were documented [13,15]. Psychosis symptoms remained stable throughout the lumateperone treatment period, as evidenced by a mean PANSS total score of 72.2 (±7.39 SD) at baseline and 73.2 at day 11 (±9.93) [15].

### 3.3. Phase III Clinical Trials

A safety/efficacy phase III double-blind, placebo-controlled, multi-center study (ITI-007-301; NCT02282761) randomized 450 adult patients (aged 18–60 years) with acute exacerbation of schizophrenia to placebo (*n* = 150), 40 mg once daily (*n* = 150) or 60 mg once daily (*n* = 150) of oral lumateperone for 4 weeks [11]. The primary outcome was the change from baseline to day 28 in the PANSS total score. The secondary outcomes were the changes in CGI-S score and other secondary efficacy measures such as PANSS subscales, CDSS, and Personal and Social Performance Scale (PSP) [11]. This study demonstrated that lumateperone at a 60 mg dosage exhibited statistical significance when compared to a placebo in terms of the change in PANSS total score from baseline to day 28, with a least squares mean difference (LSMD) of −4.2 when compared to the placebo (95% CI = −7.8 to −0.6, *p* = 0.02, effect size, −0.30). By the eighth day of the trial, statistically significant differences from the placebo were observed in the PANSS total score within the 60 mg lumateperone group. Those in the 60 mg lumateperone groups, in addition, had a statistically significant change in CGI-S score from baseline in comparison to placebo (LSMD from placebo −0–3; unadjusted *p* = 0.003). The effect of lumateperone 40 mg once daily versus placebo was not statistically relevant regarding the primary endpoint (LSMD −2.6; 95% CI = −6.2 to 1.1; effect size, −0.2; nominal *p* = 0.16); however, a notable distinction was observed in the CGI-S score when compared to the placebo (LSMD from placebo −0.2; 95% CI = −0.5 to 0; effect size: −0.3; nominal *p* = 0.02) [11]. Both treatment groups displayed significant improvements in PANSS positive symptoms from baseline to day 28 compared to the placebo (60 mg: LSMD −1.7; nominal *p* = 0.006 and 40 mg: LSMD −1.2; nominal *p* = 0.04) [11]. The modifications in the PANSS negative subscale score from the baseline to day 28, when compared to the placebo, did not show statistical significance in either group. However, statistically significant changes were detected in the general psychopathology subscale score and in psychosocial performance with the use of 60 mg of lumateperone (measured via the PANSS derived prosocial factor and PSP scale; general psychopathology subscale: LSMD, −2.4; 95% CI =−4.3 to −0.5; effect size, −0.3; nominal *p* = 0.01; PANSS-derived prosocial factor: LSMD, −1.1; 95% CI = −2.2 to 0.0; effect size, −0.2.; nominal *p* = 0.04; and PSP scale: LSMD, 3.3; 95%CI: 0.1 to 6.6; effect size,−0.3; nominal *p* = 0.05) [11]. The variation in CDSS from baseline to day 28 did not exhibit a statistically significant difference compared to the placebo group following the administration of 60 mg lumateperone (LSMD, 0.4; 95% CI = −0.24 to 0.96; nominal *p* = 0.24) or 28 mg of lumateperone (LSMD,0.2; 95% CI = −0.43 to 0.79; nominal *p* = 0.57) [11].

Regarding the drug’s safety, lumateperone exhibited a favorable side effect profile. TEAEs occurring at a rate greater than twice that of the placebo, or in at least 5% of patients exposed to lumateperone, for lumateperone 60 mg, 40 mg, and placebo, respectively, were as follows: somnolence (17%, 11.3%, and 5.4%), sedation (12.7%, 9.3%, and 5.4%), fatigue (5.3%, 4.7%, and 1.3%), and constipation (6.7%, 4%, and 2.7%). Two serious TEAEs were also assessed: one case of convulsion in the 40 mg lumateperone group and one case of orthostatic hypotension in the 60 mg lumateperone group. No serious or adverse drug effects were reported during the study. The use of lumateperone did not show any correlation with increased EPSs, ECG changes, metabolic changes, and weight changes compared with placebo [11].

In another phase III trial, conducted as a double-blind, placebo-controlled, multi-center study (ITI-007302; NCT02469155), 696 adult patients (aged 18–60 years) experiencing acute exacerbation of schizophrenia were randomized to receive either 20 mg once daily (*n* = 174) or 60 mg once daily (*n* = 174) of oral lumateperone, risperidone 4 mg once daily, or placebo for a duration of 6 weeks. The primary outcome measured was a modification of the PANSS total score from baseline to day 42 [17]. Notably, neither lumateperone dosage exhibited significant changes in PANSS total score when compared to the placebo, distinguishing these results from those of the previous trial. A high placebo response was evaluated and the investigators postulated that this was responsible for the results [17].

### 3.4. Further Studies

A pooled study was conducted on three phase II/III double-blind, placebo-controlled trials involving patients with acute exacerbation of schizophrenia (NCT00282761, NCT020469155, NCT01499563) to determinate the safety and tolerability of lumateperone 60 mg (Table 2) [40]. In two of these studies, risperidone 4 mg served as an active comparator. The pooled population included 1073 patients undergoing acute exacerbation of schizophrenia, randomly assigned to receive placebo (*n* = 412), risperidone 4 mg (*n* = 255) or lumateperone 60 mg (*n* = 406). Efficacy data were collected from the two studies (NCT020469155 and NCT01499563), while safety data were collected from all three studies. In the efficacy analyses involving 530 patients, lumateperone 60 mg significantly decreased the PANSS total score (LSMD −4.76; *p* < 0.001). The efficacy of 60 mg lumateperone was comparable to that of 4 mg risperidone (LSMD −4.97; *p* = 0.014) [40]. TEAEs were predominantly mild, and the discontinuation rate caused by TEAEs with lumateperone 60 mg (0.5%) was comparable to placebo (0.5%) and lower than with risperidone (4.7%). The only TEAEs that showed up at a rate >5% and twice that of placebo for lumateperone were somnolence/sedation and dry mouth [40]. The incidence of EPS was similar for patients treated with lumateperone 60 mg and placebo, and lower than for risperidone-treated patients [40]. This pooled analysis highlighted safety and efficacy of lumateperone 42 mg [40].

Another pooled study was carried out on the same three phase II/III, double-blind, short-term placebo-controlled trials carried out on patients with acute symptoms of schizophrenia (NCT00282761, NCT020469155, NCT01499563) to study motor symptoms and EPS [41]. The 1 year-long open-label trial, registered as NCT03817528, was also analyzed to assess the risk of EPS during long-term treatment. The evaluation of EPS encompassed the incidence of EPS-related TEAEs and the duration until their onset.

The pooled short-term safety population included 1073 patients (placebo = 412; lumateperone 60 mg = 406; risperidone 4 mg = 225). The safety analysis for the long-term open-label study involved 239 patients who successfully completed one year of treatment. The pooled short-term data demonstrated that EPS were less frequent in patients receiving 60 mg lumateperone (3.0%) or placebo (3.2%) compared with those patients who received risperidone 4 mg (4.9%) [41]. Among the EPS, akathisia was the most frequently described following lumateperone intake. Regarding the time to onset of EPS, side effects appeared to manifest earlier in risperidone (9 days) than in either placebo (14 days) or lumateperone (17 days). Long-term treatment with lumateperone was correlated with a low incidence of EPS-related TEAE (0.8%) and the time to EPS symptom onset was 38 days [41]. 

### 3.5. Ongoing Studies

Study NCT04959032/ITI-007-304 is a multi-center, randomized, double-blind, phase III placebo-controlled, parallel-group, fixed-dose study conducted in 200 adult patients (aged 18–60 years) with a diagnosis of schizophrenia [42]. The primary outcome is to evaluate the time to first symptom relapse through the double-blind treatment. The study started in July 2021 and is expected to finish in December 2023. The study will be conducted in five phases: an initial no-drug Screening Phase lasting up to 7 days, during which patient eligibility will be assessed; a 6-week open-label Run-in Phase (RIP), where all patients will be administered oral lumateperone at a dosage of 42 mg/day; a subsequent 12-week open-label Stabilization Phase (SP), during which all patients will continue to receive oral lumateperone at a dosage of 42 mg/day; a 26-week Double-blind Treatment Phase (DBTP), wherein patients will be randomly assigned to receive either lumateperone 42 mg or a placebo (in a 1:1 ratio); and, finally, a 2-week Safety Follow-up (SFU) Phase [42]. Currently, no data are available from this study. Moreover, this is the only on-going study for the treatment of schizophrenia that uses lumateperone [42].

## 4. Discussion

Lumateperone is a novel second-generation antipsychotic with efficacy for schizophrenia and with an advantageous adverse event profile evaluated in both short- and long-term tolerability studies [11,15,17,26,38,39]. Its efficacy as an atypical antipsychotic drug is derived from a combination of presynaptic partial agonism and postsynaptic antagonism at central dopamine D2 receptors and antagonist activity at the serotonin 5HT2A receptors [16,43,44,45]. In particular, the abovementioned D2 receptor activity shows a regional selectivity for both mesocortical and mesolimbic circuits with a low affinity for nigrostriatal dopamine pathways [19,46]. In addition, lumateperone is a receptor-dependent modulator of glutamate, and a serotonin reuptake inhibitor [1]. Moreover, it has been hypothesized that lumateperone may have a reduced capability in determining EPS and hyperprolactinemia because of its binding affinity for 5HT2A receptors, which is 60 times higher than for D2 receptors, its unique interaction with D2 receptors at the synaptic level with functional mesolimbic and mesocortical selectivity, and its 40% striatal D2 receptor occupancy (D2RO) at peak plasma concentrations, which is a lower occupancy compared to other antipsychotics [17].

Lumateperone has been demonstrated to be efficacious in the management of positive, negative, and cognitive symptoms related to schizophrenia [2,3,47]. The findings from the Phase II/III clinical trials (NCT01499563/ITI-007-005; NCT02282761/ITI-007-301) with patients suffering from an exacerbation of schizophrenia indicate that lumateperone 60 mg has antipsychotic properties superior to placebo [11,26]. In the NCT01499563 phase II, placebo-controlled study, a significant improvement from baseline to day 28 was observed in the PANSS total score for the 60 mg lumateperone group [26]. Furthermore, the phase II study with the identifier NCT02282761, which employed a placebo control, indicated the statistical significance of lumateperone 60 mg compared to the placebo in terms of the change in PANSS total score from baseline to day 28, with a Least Squares Mean Difference (LSMD) relative to placebo of −4.2 [11]. However, the first study did not demonstrate the efficacy of 120 mg lumateperone, which resulted in the improvement of the PANSS total score not being statistically significant [11]. The causes of the lack of efficacy of the 120 mg dose of lumateperone are not yet known. It appears plausible that some patients administered a 120 mg dose of lumateperone may experience improvements in specific PANSS items while displaying deteriorations in other items. This could lead to an overall effect of no change in the total PANSS score when compared to the baseline [14]. In addition, it has been argued that the mixed effect of presynaptic D2 agonism/postsynaptic D2 antagonism could explain why the higher dose of lumateperone is less effective [14]. In the second study, however, patients treated with 40 mg did not show a statistically significant decrease in the PANSS total score. Nonetheless, there was a decrease reported in the PANSS positive symptoms subscale and the CGI-S [11]. In both of the ITI-007-005 and ITI-007-301 studies, there were not statistically significant enhancement in the PANSS negative subscale at day 28 compared to placebo in all the three lumateperone treatment subgroups (60 mg lumateperone, 120 mg lumateperone, and 40 mg lumateperone) [11,26]. Indeed, negative symptoms only ameliorated with 60 mg lumateperone in study ITI-007-005; however, this was not statistically significant [26].

Enhancing psychosocial outcomes is a frequently pursued yet frequently unattained goal for individuals with schizophrenia [48]. Improvements in psychosocial function in study ITI-007-005 and ITI-007-301 were suggested by improvements, in the lumateperone 60 mg group, in PANSS-derived prosocial factor score and PANSS general psychopathology subscale score [11,26].

Despite the favorable outcomes observed in the initial two trials, the NCT02469155/ITI-007-302 trial revealed that following a 6-week treatment for schizophrenia, lumateperone doses of 20 mg and 60 mg did not demonstrate efficacy, as the results were comparable to those of the placebo; the investigators postulate that this was due to the high placebo response rate [17]. Studies ITI-007-005 and ITI-007-302 included an active comparator arm (risperidone 4 mg once daily). In study ITI-007-005, both risperidone 4 mg and lumateperone 60 mg were found to be effective compared with placebo. In study ITI-007-302, which enrolled a larger study population and had an extended duration compared to the other placebo-controlled trial, only the 4 mg dose of risperidone showed a significant difference compared to the placebo [40]. In studies ITI-007-005, ITI-007-301, and ITI-007-302, the majority of treatment discontinuation was caused by the inadequate efficacy of the 20 mg and 40 mg doses of lumateperone compared to the 60 mg dose. Additionally, in study ITI-007-302, discontinuation due to lack of efficacy was more frequent for the 20 mg dose of lumateperone than for the placebo. Across all three placebo-controlled studies, the discontinuation of treatment due to insufficient efficacy was more prevalent in the placebo group when compared to the lumateperone 60 mg group, demonstrating the effectiveness of the treatment [40]. Because of the lack of efficacy of diverse doses, lumateperone is currently approved by FDA for the treatments of adult patients affected by schizophrenia at a dose of 60 mg of lumateperone tosylate (42 mg of active moiety) once daily [5].

Lumateperone has also proved its efficacy when in the phase II, open-label trial ITI-007-303/NCT03817528, conducted in outpatients with stable schizophrenia, when switching from previous antipsychotics to 6 weeks of lumateperone 60 mg treatment, patients remained stable or still had an improvement in symptoms, as evidence by mean improvements in the PANSS total score [17]. This is also the only long-term trial already conducted. In fact, patients were observed for changes in the PANSS total score for 1 year. Lumateperone led to a reduction in schizophrenic symptoms as shown by the significant reduction in the PANSS total score as compared to baseline [39].

Furthermore, the results of an open-label, non-randomized phase I Clinical Trial (ITI-007-025; NCT04709224), conducted to evaluate the safety, pharmacokinetics and tolerability of single doses of a subcutaneous long-acting injection of lumateperone that increase progressively, are still being processed [36]. This type of administration could help in the management of patients with low compliance, reducing the side effects that may arise from using other long-acting injectables (LAIs) due to its tolerability profile, which often results in treatment discontinuation [49]. However, additional data are required.

According to the safety parameters considered in several clinical trials, lumateperone is considered a safe and well-tolerated antipsychotic. [11,17,39] At the FDA-approved dosage of 60 mg/day, the most frequent side effects are somnolence, sedation, fatigue, and constipation [5].

Lumateperone has a low metabolic risk and causes minimal weight gain in comparison to commonly used second-generation antipsychotics [1,5,43]. No relevant significant variations in metabolic parameters, ECG abnormalities (e.g., QTc prolongation) and prolactin levels were observed after lumateperone treatment [16,44]. Additionally, there is a decreased occurrence of EPS [17]. Lumateperone causes minimal elevation of serum aminotransferase throughout the duration of the therapy, but no clinically evident acute liver injury related to its use has been observed [50].

In the phase II/III placebo-controlled trials (ITI-007-005; ITI-007-301; ITI-007-302), the most frequent adverse events observed with lumateperone 60 mg (LUM) included nausea (9% LUM vs. 5% PBO), somnolence/sedation (24% LUM vs. 10% placebo (PBO)), dizziness (5% LUM vs. 3% PBO), hepatic enzymes increased (2% LUM vs. 1% PBO), dry mouth (6% LUM vs. 2% PBO), fatigue (3% LUM vs. 1% PBO), increased creatine phosphokinase (4% LUM vs. 1% PBO), and vomiting (3% LUM vs. 2% PBO) [40]. Lumateperone did not demonstrate a substantial risk of inducing EPS when compared to the placebo. During the 6-week study ITI-007-302, only two patients who were administered lumateperone 60 mg reported tardive dyskinesia [17]. Tardive dyskinesia, as defined, is an extrapyramidal symptom that emerges later in the course of prolonged antipsychotic treatment. [27]. It is extremely unlikely for a new antipsychotic to induce tardive dyskinesia within the first 6 weeks of treatment [27]; it is more likely that the observed cases of tardive dyskinesia were, in fact, manifestations of withdrawal dyskinesia.

Withdrawal dyskinesia might exhibit similarities to tardive dyskinesia upon evaluation and can occur in individuals who switch antipsychotic medications or cease long-term antipsychotic treatment [51].

In an open-label investigation (ITI-007-303), lumateperone exhibited minimal metabolic, EPS, and cardiovascular safety concerns when contrasted with the current SOC antipsychotic therapy. Tardive dyskinesia emerged as a treatment-related event in only one patient (0.2%) receiving lumateperone 42 mg [39]. In this study, the serious adverse event that led to most cases of discontinuation of drug administration was the worsening of symptoms correlated with schizophrenia. However, it is not rare for individuals affected by schizophrenia to undergo periods of illness exacerbation within a one-year timeframe, and it is plausible that a significant portion of these patients may not have fully adhered to the prescribed study medication [39].

Lumateperone is not approved for the treatment of dementia-related psychosis because, as all antipsychotic medication, it increases the risk of death for this kind of patients [52]. Only hypersensitivity to lumateperone is a contraindication to prescription.

Lumateperone undergoes metabolism primarily through the cytochrome P450-3A4 isozyme and exhibits interactions with medications that function as inhibitors and/or inducers of this isozyme [50]. Consequently, it is advised to refrain from prescribing lumateperone to patients concurrently receiving CYP3A4 inducers or inhibitors. Patients on treatment with lumateperone have an enhanced fall risk due to postural hypotension or somnolence; thus, monitoring is recommended [52].

### Limitation

The reliability of currently available data for lumateperone is hampered by the pharmaceutical company that has an economical interest in marketing the new drug. Indeed, there are very few studies conducted in clinical practice that are not sponsored by the producing company. Therefore, it will be useful to conduct a larger systematic review when data from different clinical centers without financial interests will be available. In addition, future reviews will also include the post-marketing surveillance for adverse effects.

## 5. Conclusions

Schizophrenia is an intricate and severe mental disorder that disrupts the normal processing of information in the brain. The standard of care when treating schizophrenia is pharmacological; antipsychotics are the only adequate drug for the treatment of this disease. Historically, antipsychotics have had numerous side effects, particularly extrapyramidal side effects and metabolic issues.

First-generation antipsychotic drugs indiscriminately block dopamine D2 receptors, and the non-selective nature of these drugs can result in various side effects for patients. In contrast, second-generation antipsychotic drugs selectively block dopamine D2 receptors and serotonin 5-HT2A receptors, making them the most commonly prescribed medications for the treatment of schizophrenia. Lumateperone is a first-in-class, small molecule, new molecular entity that selectively and simultaneously modulates serotonin, dopamine, and glutamate neurotransmission. The whole of this peculiar pharmacodynamic, pharmacokinetic, and safety profile of lumateperone led to it being approved by the U.S FDA in December 2019 to treat schizophrenia in adult patients. It may be considered a safe antipsychotic drug due its good side effects profile. Studies have demonstrated that the most frequent TEAEs observed during pharmacological therapy with lumateperone are dry mouth, nausea, dizziness and somnolence/sedation. Furthermore, there is a low risk of hyperglycemia, weight gain, hyperprolactinemia, dyslipidemia and EPS compared to other drugs approved for the treatment of schizophrenia and currently utilized as the standard of care. However, further data are required before a definitive evaluation regarding the effectiveness and safety of lumateperone, particularly when considering its use for the extended treatment of schizophrenia. Additional research, which can assess the efficacy of lumateperone in comparison to existing antipsychotic medications, should be conducted, especially in relation to the most recent antipsychotic options. Side effects should be evaluated in post-marketing studies to demonstrate whether lumateperone may be an antipsychotic safer than the other second-generation antipsychotics. One of the next steps should involve the opportunity of investigating the effects of long-acting lumateperone in a more extensive cohort of participants and to compare the efficacy and safety of lumateperone LAI with the ones already on the market. In conclusion, lumateperone is currently being explored in clinical trials for both major depressive disorder and behavioral agitation related to Alzheimer’s disease and other dementias.

## Figures and Tables

**Figure 1 brainsci-13-01641-f001:**
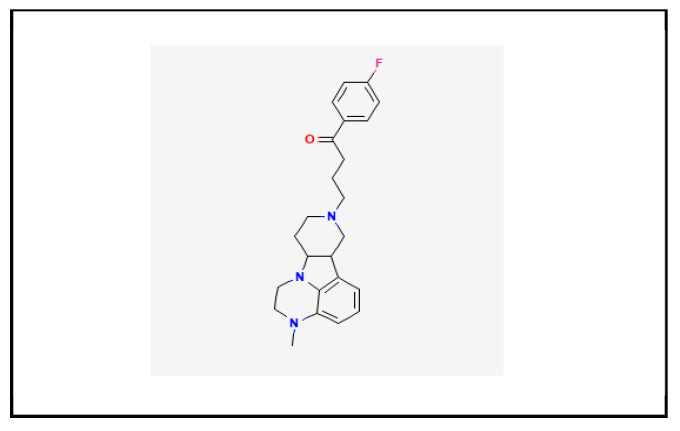
Chemical structure of lumateperone.

**Figure 2 brainsci-13-01641-f002:**
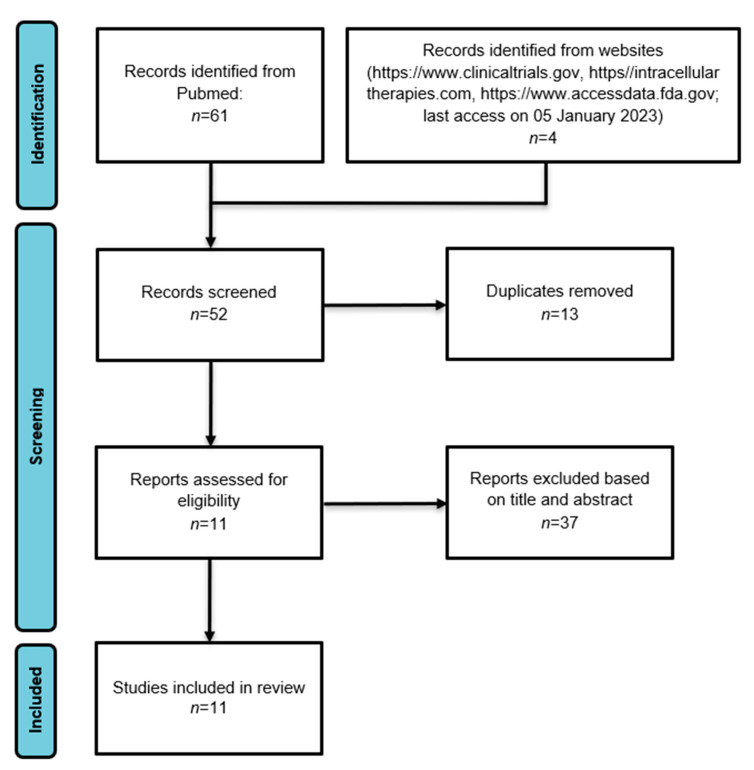
Flowchart illustrating the process of study search and selection in accordance with PRISMA guidelines.

**Table 1 brainsci-13-01641-t001:** Lumateperone and schizophrenia: summary of clinical trials.

Clinical Trials.gov ID Number (Current Status) Period of Study	Peer-Reviewed Publication and/or Abstract Conference	Study Design	Endpoint Classification	Aim/Hypothesis	Sample Characteristics	Study Description	Outcomes		Adverse Effects Observed during the Studies
							Primary	Secondary	
NCT04709224 ITI-007-025 (completed) November 2021–December 2021.	Unpublished [36]	Open-label study-Phase I.	Pharmacokinetics/Safety/Tolerability.	Study examining the safety, pharmacokinetics, and tolerability of progressively increasing doses of the long-acting injectable formulation of lumateperone in adults diagnosed with schizophrenia, who are clinically stable and have been free from acute exacerbation for a minimum of 3 months.	*n* = 37. Age: 18–50 years. Diagnosis: schizophrenia diagnosis, clinically stable.	Length: 4 wks. double-blind treatment. Experimental: LAI lumateperone 50 mg subcutaneous on the abdomen, 100 mg subcutaneous on the abdomen, 200 mg subcutaneous on the abdomen, 100–200 mg subcutaneous in the outer area of the upper arm	Cmax Tmax AUC T ½	Safety/ Tolerability as Measured by Number of Participants with AEs; Change from baseline in: - systolic and diastolic blood pressure; - platelet count; - white blood cell count; - alanine aminotransferase - hemoglobin; - aspartate aminotransferase; - glucose; - creatine kinase; - ECG QTc interval.	Data not available
ITI-007-020; NCT04779177 (completed) March 2021–December 2022.	Unpublished [37]	Open-label study-Phase I.	Pharmacokinetics/Safety/Tolerability.	Study assessing the pharmacokinetics, safety, and tolerability of a single oral dose of lumateperone in adolescent patients diagnosed with schizophrenia or schizoaffective disorder, who are clinically stable and have been free from acute exacerbation for at least 3 months.	*n* = 26. Age: 13–17 yy. Diagnosis: schizophrenia and schizoaffective disorder diagnosis, clinically stable.	Length: 30 days. Experimental: 28 mg/d, 42 mg/d	Cmax Tmax AUC T ½ CL/F	Safety/ Tolerability as Measured by Number of subjects with treatment-emergent adverse events. Change from baseline in: - systolic and diastolic blood pressure; - white blood cell count; - alanine aminotransferase; - aspartate aminotransferase; - hemoglobin; - ECG QTc interval; - Abnormal Involuntary Movement Scale (AIMS).	Data not available
NCT01499563 ITI-007-005 (completed) December 2011–November 2013.	[26]	Double-blind, placebo-controlled, randomized, study-Phase II.	Safety/Efficacy study.	Study investigating the safety and effectiveness of lumateperone in adult patients undergoing an acute exacerbation of schizophrenia.	*n* = 335. Age: 18–55 y. Diagnosis: schizophrenia diagnosis, current psychotic episode.	Length: 4 wks. double-blind treatment. Experimental: 60 mg/d (*n* = 76), 120 mg/d (*n* = 80) of oral lumateperone. Comparator: placebo (*n* = 80) orally. Active comparator: risperidone 4 mg/d (*n* = 75)	PANSS	PANSS subscales CDSS.	Dry mouth, worsening of schizophrenia, somnolence/sedation, nausea, dizziness.
NCT03817528 ITI-007-303a (completed) March 2019–March 2021.	[17,38]	Open-label study-Phase II.	Safety/Efficacy study.	Study exploring efficacy and safety of lumateperone in adults patients affected by stable schizophrenia, who were switched from SOC.	*n* = 302. Age: 18–60. Diagnosis: schizophrenia diagnosis, stable.	Length: 6 wks. Fw-up: 40 wks. Experimental: lumateperone 60 mg once daily.	TEAEs	PANSS.	Somnolence, dry mouth, headache, migraine, panic attack, hepatic enzyme augmentation, non-cardiac chest pain.
NCT03817528 ITI-007-303b Continuation of study ITI-007-303a.	[39]	Open-label study-Phase II.	Safety/Efficacy study.	Study exploring efficacy and safety of lumateperone in adults patients affected by stable schizophrenia, who were switched from SOC.	*n*= 603. Age: 18–60. Diagnosis: schizophrenia diagnosis, stable.	Length: 120 wk. Experimental: lumateperone 60 mg once daily.	TEAEs	PANSS CDSS.	Anxiety, suicidal ideation, EPS, dizziness, diarrhea, nausea, vomiting, insomnia, fatigue.
NCT02288845 ITI-007-008 (completed) October 2014–September 2015.	[15]	Open-label study-Phase II.	Receptor Occupancy/Safety/ Tolerability and Pharmacokinetics	Study evaluating D2 receptor occupancy, tolerability and safety of lumateperone in adult patients affected by schizophrenia in clinical remission.	*n* = 14. Age:18–60 yy. Diagnosis: schizophrenia diagnosis, in clinical remission and free from acute exacerbation of psychosis.	Length: 2 wks. Experimental: 60 mg/d of oral lumateperone.	Brain Receptor Occupancy as Measured by PET.	Safety/ Tolerability as Measured by Number of Participants with AEs.	Mild to moderate headache, mild sedation.
NCT0228276 ITI-007-301 (completed) November 2014– September 2015.	[11]	Double-blind, placebo-controlled, randomized, study-Phase III.	Safety/Efficacy study.	Study assessing the safety and effectiveness of lumateperone in adult patients going through an acute exacerbation of schizophrenia.	*n* = 450. Age: 18–60 yy. Diagnosis: schizophrenia diagnosis, current psychotic episode.	Length: 4 wks. double-blind treatment. Experimental: 40 mg/d (*n* = 150), 60 mg/d (*n* = 150) of oral lumateperone. Comparator: placebo (*n* = 150) orally.	PANSS	PANSS subscales; CGI-S, CDSS, PSP.	Somnolence, sedation, fatigue, constipation, headache, nausea, dry mouth, dizziness.
NCT02469155 ITI-007-302 (completed) June 2015– August 2016.	[17]	Double-blind, placebo-controlled, randomized with active comparator, study-Phase III	Efficacy study	Study evaluating the efficacy of lumateperone in adult patients experiencing an acute exacerbation of schizophrenia.	*n* = 696. Age: 18–60 yy. Diagnosis: schizophrenia diagnosis, current psychotic episode.	Length: 6 wks. double-blind treatment. Experimental: 20 mg/d (*n* = 174), 60 mg/d (*n* = 174) of oral lumateperone. Comparator: placebo (*n* = 174) orally. Active comparator: risperidone 4 mg/d (*n* = 174)	PANSS	PANSS subscales.	Somnolence, dry mouth and headache.

**Note:** Sample number= *n*; years = yy; weeks = wks; day = d; Positive and Negative Symptoms Scale = PANNS; Clinical Global Impression-Improvement Scale = CGI; Long Active Injection = LAI; Treatment-Emergent Adverse Events = TEAEs; Calgary Depression Scale for Schizophrenia = CDSS; AEs = adverse events; Personal and Social Performance Scale = PSP; Follow-up period: Fw-up; Standard of Care = SOC; Positron Emission Tomography = PET; Maximum observed plasma concentration = Cmax; Time of maximum observed plasma concentration = Tmax; Area under the plasma concentration-time curve = AUC; Terminal Elimination half-life = T½.

**Table 2 brainsci-13-01641-t002:** Lumateperone and schizophrenia: summary of further studies included.

References	Study Design	Study Description
[40]	Pooled analysis: NCT00282761 NCT02469155 NCT01499563	A pooled analysis of three phase II/III clinical trials carried out on individuals with schizophrenia to assess the safety and efficacy of lumateperone.
[41]	Pooled analysis: NCT00282761 NCT02469155 NCT01499563 NCT03817528 (b)	A pooled analysis of four phase II/III clinical trials carried out on individuals with schizophrenia in order to evaluate motor symptoms and EPS.

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
