# Peer review of "The Novel Antipsychotic Lumateperone (Iti-007) in the Treatment of Schizophrenia: A Systematic Review"

_brainsci, 2023, doi:10.3390/brainsci13121641_

Round 1
Reviewer 1 Report
Comments and Suggestions for Authors
Objectives? - There is a review on this topic published in 2021, objectives should be clearly outlined and it should be explained why authors decided to repeat the review
More details on methodology required - inclusion/exclusion criteria, how many researchers screened the abstracts/full texts, how were conflicts resolved? Why was the search conducted only on PubMed?
Limitations and strengths of the study should also be reported in detail
Comments on the Quality of English Language
Language refinement advised
Author Response
- Objectives? - There is a review on this topic published in 2021, objectives should be clearly outlined and it should be explained why authors decided to repeat the review.
Dear reviewer, we perfectly agree with your advice. According to your suggestions and annotated comments we have coherently and thoroughly revised our manuscript, by clearly specifying objectives.
- More details on methodology required - inclusion/exclusion criteria, how many researchers screened the abstracts/full texts, how were conflicts resolved? Why was the search conducted only on PubMed?
Dear reviewer, we perfectly agree with your advice. According to your suggestions and annotated comments we have coherently and thoroughly revised our manuscript. We specified this issue in the Materials and Methods section of our manuscript.
- Limitations and strengths of the study should also be reported in detail.
Dear reviewer, many thanks for your valuable comments.
According to your suggestions and annotated comments we have coherently and thoroughly revised our manuscript, by clearly specifying limitations and strengths in the Discussion section.
Reviewer 2 Report
Comments and Suggestions for Authors
In this article, entitled “NEW ANTIPSYCHOTIC LUMATEPERONE (ITI-007) IN THE TREATMENT OF SCHIZOPHRENIA,” author Longo and colleagues delve into the description of a new antipsychotic drug approved for use by the FDA in 2019. This review article is certainly of great importance for practicing psychiatrists because it summarizes the available data on the pharmacodynamics, pharmacokinetics, effectiveness and safety of the new drug.
I would like more information about the safety of the drug. In the "Preclinical Studies" section, please expand the information on the toxicity of LUMATEPERONE. Add LD50 and NOAEL values if determined.
I would like to see in supplementary a summary table of AEs obtained in clinical studies.
The numbering of the results is not entirely clear. Section 3.1 is entitled Lumateperone in Schizophrenia. But all the results in this section are presented only for patients with schizophrenia. There are no studies of lumateperone in other psychiatric diseases. I propose to either remove the numbering or add information on studies of lumateperone in depressive and bipolar disorders as indicated in the introduction.
In general, this review article seems very interesting and informative for a wide range of readers.
Best regards,
Reviewer
Author Response
- I would like more information about the safety of the drug. In the "Preclinical Studies" section, please expand the information on the toxicity of LUMATEPERONE. Add LD50 and NOAEL values if determined.
Dear reviewer, many thanks with your valuable comments. According to your advice, we have coherently and thoroughly revised our manuscript by adding NOAL information. Unfortunately, LD50 was not determined in preclinical studies.
- I would like to see in supplementary a summary table of AEs obtained in clinical studies.
Dear reviewer, many thanks with your valuable comments. According to your suggestions and annotated comments we have included the AEs of clinical studies in Table 1.
- The numbering of the results is not entirely clear. Section 3.1 is entitled Lumateperone in Schizophrenia. But all the results in this section are presented only for patients with schizophrenia. There are no studies of lumateperone in other psychiatric diseases. I propose to either remove the numbering or add information on studies of lumateperone in depressive and bipolar disorders as indicated in the introduction.
Dear reviewer, we perfectly agree with your advice. Accordingly, we have properly corrected the numbering of the section of our manuscript.
Reviewer 3 Report
Comments and Suggestions for Authors
In this paper, Longo and colleagues provide a comprehensive overview of the available evidence regarding the safety, tolerability, and effectiveness of new second-generation antipsychotic lumateperone in the treatment of schizophrenia. They provide a thorough pre-clinical background and synthesise the existing evidence from available phase I, II, and III trials, also taking care of preliminary data from ongoing studies.
The manuscript is very straightforward and detailed from a conceptual point of view, offering a precious up-to-date review of the evidence available about this brand new pharmacological option for the management of schizophrenia, and surely represents an important contribution to the scientific literature concerning the treatment of schizophrenia.
Nonetheless, while the Results section seems to suggest that this work systematically reviews all existing studies about lumateperone in schizophrenia, the Authors do not declare the study as such (generally talking about an “overview”) and the Methods do not point towards a systematic approach.
I suggest that the Authors follow the PRISMA statement, make the type of paper explicit in the title as well as in the abstract, update the Methods section, provide a PRISMA flowchart, and include Table S1 as a main table (not relegating it as a supplement).
Some more minor comments:
1. Abstract: the Abstract is a bit unbalanced, providing too much background at the expenses of the findings of this review. The Authors should add a brief summary of the evidence they have synthesised.
2. How might lumateperone fit into the current landscape of antipsychotics on the market? Please refer and compare to relevant literature on the topic (e.g., https://doi.org/10.2147/prbm.s371991). The Authors should discuss this issue in the Discussion and Conclusions sections.
3. The Authors should add a brief statement about the possible limitations of their overview.
Author Response
- Nonetheless, while the Results section seems to suggest that this work systematically reviews all existing studies about lumateperone in schizophrenia, the Authors do not declare the study as such (generally talking about an “overview”) and the Methods do not point towards a systematic approach.
Dear reviewer, we perfectly agree with your advice. According to your suggestions and annotated comments we have coherently and thoroughly revised our manuscript by reorganizing Materials and Methods and Results sections.
- I suggest that the Authors follow the PRISMA statement, make the type of paper explicit in the title as well as in the abstract, update the Methods section, provide a PRISMA flowchart, and include Table S1 as a main table (not relegating it as a supplement).
Dear reviewer, we perfectly agree with your advice. According to your suggestions and annotated comments we have coherently and thoroughly revised our manuscript by reporting the PRIMA flowchart.
- Abstract: the Abstract is a bit unbalanced, providing too much background at the expenses of the findings of this review. The Authors should add a brief summary of the evidence they have synthesised.
Dear reviewer, many thanks with your valuable comments. According to your suggestions and annotated comments we have coherently and thoroughly revised the abstract.
- How might lumateperone fit into the current landscape of antipsychotics on the market? Please refer and compare to relevant literature on the topic (e.g., https://doi.org/10.2147/prbm.s371991). The Authors should discuss this issue in the Discussion and Conclusions sections.
Dear reviewer, we perfectly agree with your advice. Accordingly, we have properly included the above mentioned reference within the main text in the Discussion section (page 15) as well as in the reference section.
- The Authors should add a brief statement about the possible limitations of their overview.
Dear reviewer, many thanks with your valuable comments. According to your suggestions and annotated comments we have coherently and thoroughly revised our manuscript by clearly specifying limitations in the Discussion section.
Round 2
Reviewer 3 Report
Comments and Suggestions for Authors
The Authors put a commendable effort in methodologically systematizing their article, which is now much improved and way more solid.
Few further observations:
1. The number of records found though the systematic search and the number of papers ultimately included in the review represent a result and not a method. Thus, they should be moved at the very beginning of the Results section of the manuscript.
2. The PRISMA flowchart can be improved, also considering that the numbers do not sum up.
Author Response
- The number of records found though the systematic search and the number of papers ultimately included in the review represent a result and not a method. Thus, they should be moved at the very beginning of the Results section of the manuscript.
Dear reviewer, we perfectly agree with your advice. According to your suggestions and annotated comments we have coherently and thoroughly revised our manuscript by reorganizing Materials and Methods and Results sections.
- The PRISMA flowchart can be improved, also considering that the numbers do not sum up.
Dear reviewer, many thanks with your valuable comments. According to your suggestions and annotated comments we have coherently and thoroughly revised the PRISMA flowchart.